# Emotional State of Teachers and University Administrative Staff in the Return to Face-to-Face Mode

**DOI:** 10.3390/bs12110420

**Published:** 2022-10-30

**Authors:** Hugo Arias-Flores, Jorge Guadalupe-Lanas, Doris Pérez-Vega, Verónica Artola-Jarrín, Jorge Cruz-Cárdenas

**Affiliations:** 1Centro de Investigación en Mecatrónica y Sistemas Interactivos—MIST, Universidad Tecnológica Indoamérica, Quito 170103, Ecuador; 2Independent Researcher, Quito 170103, Ecuador; 3Facultad de Ciencias Humanas y de la Salud, Universidad Tecnológica Indoamérica, Quito 170103, Ecuador; 4Facultad de Ciencias Económicas, Pontificia Universidad Católica del Ecuador, Quito 170143, Ecuador; 5Research Center in Bussiness, Society and Technology (ESTec), Facultad de Ciencias Administrativas, Universidad Tecnológica Indoamérica, Quito 170103, Ecuador

**Keywords:** administrative staff, COVID-19, DASS-21, face-to-face education, teachers

## Abstract

Social distancing and security measures have contained the spread of the COVID-19 pandemic. Despite this, the return to face-to-face activities is necessary for specific companies, and some higher education institutions have already done so. The various disorders that this new reality could generate have motivated the present study, which aims to analyze the emotional state of teachers and administrative staff. The instrument used was the abbreviated depression, anxiety, and stress scale (DASS-21), with an internal consistency index of 0.87. The methodology was based on applying a survey to 202 participants from Quito, Ecuador. The sample consisted of 97 men and 105 women aged between 23 and 59 years. A quantitative and cross-sectional design was used in this research. The results show that 40.1% of the respondents presented anxiety, 36.63% depression, and 38.61% stress between mild and highly severe categories. Additionally, when analyzing the depression, anxiety, and stress levels compared to productivity variables, we found that five disorders, i.e., fear, anxiety, over reactivity, skeletal muscle effects, and dysphoria, directly affect productivity variables, such as performing simple tasks, performing difficult tasks, the number of products made, and the number of products rejected. Thus, returning to face-to-face mode has affected the emotional state of many people, showing differences according to the job position, with anxiety being the highest self-identified incidence rate.

## 1. Introduction

Since the beginning of the COVID-19 pandemic in the year 2020, changes in human life have consisted of social distancing, the use of face masks, and partial or total confinement of the population, being the most useful [1]. In addition, some systems had to adapt to the situation, as in the case of education, which transitioned to a virtual setting. Authorities worldwide have decreed that students at all levels incorporate remote education from home. As part of this new reality, teachers have had to incorporate technological tools that have raised stress levels in those who did not use multimedia tools regularly [2]. Some higher education institutions still maintain this remote arrangement, while others have gradually reopened. Thus, students, faculty, and administrative staff have had to return to in-person activities. However, the inherent risk of contagion using protective equipment and biosecurity measures generates physical, emotional, and psychological exhaustion [3]. Since the World Health Organization (WHO) states that “health” includes the physical and mental well-being of the person, the constant assessment of their psychological and cognitive capacities should not be neglected [4]. According to Arias Gallegos et al., about burnout syndrome [5], university-level educators are the most affected by stress compared to lower levels of education, which could cause low performance, occupational illness, and absenteeism.

The changes produced by COVID-19 have had repercussions on human mental health, producing discomfort and mental disorders. This change has motivated studies to determine the influence of various factors in the development of virtual classes, as seen in Jelińska, M et al. [6]. These conditions can generate anxiety, low productivity, and difficulty in communication and attention. A study was conducted by Khan, A.H. et al., using DASS-21 and IES-R to evaluate the disorders generated in a sample of 505 university students [7]. Similarly, in Rosenthal, L. et al., the DASS-21 and IES-R instruments were applied to 222 nursing students, obtaining a high incidence of negative emotional states [8]. Similar results were found in Fawaz, M et al., with a sample of 520 university students [9].

It is also essential to analyze these disruptions in the other actors of higher education institutions. The return to classes has gradually taken place, and negative feelings have been generated amongst the teaching staff. In Ozamiz-Etxebarria, N et al., the different effects of face-to-face activities for Spanish teachers are analyzed using the DASS-21 [10]. Similarly, in Ozamiz-Etxebarria, N et al., quantifying the levels of depression, anxiety, and stress teachers suffer in a scenario of a face-to-face return to class is proposed [11].

In this context, it should be recognized that face-to-face academic activities symbolize an uncomfortable situation for the entire university ecosystem. Given that most research focuses on students and a few on teachers, this paper aims to evaluate the subject by focusing on teachers and administrative staff. Determining the levels of depression, anxiety, and stress they suffer using the DASS-21 instrument is proposed. Our hypothesis proposed that the return to classes has caused depression, anxiety, and stress, affecting people differently depending on their job position: teachers and administrative staff. Likewise, the depression, anxiety, and stress levels have affected the productivity levels measured by the numbers of products generated throughout the return to class period.

## 2. Materials and Methods

This study was carried out at the Universidad Tecnológica Indoamérica, located in the metropolitan district of Quito, where the overall quantity of workers, including university professors and administrative workers, is 300, of which 202 (67%) were taken as a representative sample, including 110 university professors and 92 administrative workers. Table 1 summarizes the information of the participants chosen in search of numerical gender equality and who voluntarily decided to be part of this research. The age range is between 23 and 59 years, where 47% are men, of which 18% are administrative, and 29% are teachers. In contrast, women represent 53%, of which 29% are administrative and 24% are teachers.

A distinction by age range shows a high number of female administrative personnel (54.17%) and a low number of male teachers (4.17%) in the category of less than 30-years-old. Moreover, in the age range between 40 to 49 years, male teachers correspond to 40.35%, and there is a small number of female administrative personnel (19.30%). For the 30 to 39 and over 50 age ranges, similar values are maintained in the distribution.

### 2.1. Procedure

The questionary was raised through the Microsoft forms platform. The participants were chosen using a non-probability sampling strategy. When statistically calculating the sample, with a confidence interval of 95% and an error of 5%, the result was 169 people to be surveyed, however, given the good opening of the participants, we exceeded 33 people as the calculated size. Confidentiality and anonymity were ensured.

### 2.2. Measures, Instruments and Data Analysis

An online questionnaire was designed and shared with potential participants to determine their primary demographic data (sex and age). In addition, the questionnaire asked whether they were carrying out face-to-face or remote activities, with an exclusion criterion for those who were not attending their institution. The questionnaire consisted of 50 questions related to socioeconomic conditions, productivity results, and the Spanish version of the Depression Anxiety and Stress Scale-21 (DASS-21), whose internal consistency index is 0.87 [12]. The questionary predominantly consisted of three sections. The first section covered demographic and institutional data (age, gender, job position, workplace, and others). The second section consisted of productivity questions related to the number of products made, time dedicated to performance activities, and the number of products rejected. The third section consisted of the Depression, Anxiety, and Stress Scale, which is a 21-item and tests three related negative emotional states (depression, anxiety, and stress). The depression scale evaluates dysphoria, hopelessness, absence of interest/involvement, devaluation of life, self-deprecation, anhedonia, and inertia. The anxiety scale judges autonomic arousal, situational anxiety, skeletal muscle effects, and experiences of anxiousness. The stress scale is receptive to persistent, non-specific arousal. It evaluates difficulty unwinding, nervous arousal, and mood changes (distress, agitation, irritability, over-reactiveness, and impatience). The DASS-21 questionnaire has a rating scale from 0 to 4, where 3 indicates more likely to occur and 0 indicates that it is not likely to occur. This instrument makes it possible to identify a tendency to certain conditions associated with psychological and physical changes in human beings.

### 2.3. Statistical Analysis

Data was collected through an online questionnaire using the Microsoft forms platform. Once information was collected, the data were exported to IBM SPSS Statistics version 26 statistical software. In order to ensure the quality of the information, we proceeded to calculate data for the missing cells using the method of the average of the nearest cells. The chi-square test was applied to the categorical data related to the DASS-21 questionnaire, while the analysis of productivity and COVID-19 was done via ANOVA and Tukey’s Honest Significant Difference test. The tests were two-tailed, and the significance level chosen was a *p*-value < 0.05. Descriptive frequencies were also applied to assess some sociodemographic variables and highlight notable trends.

## 3. Results

### 3.1. DASS-21 and Socio-Demographic Variables

Contact with potential participants was made, in some cases, verbally and in others by e-mail. After applying the exclusion criteria, a link was sent so that they could access the initial online questionnaire. They were also informed about the discreet treatment of the data collected to protect confidential information. Subsequently, the applied instrument was socialized, explaining its main characteristics and executing an example. Moreover, Table 2 shows the number of people who suffer from these disorders concerning the defined socio-demographic variables.

When analyzing the results by age range, there are 36.6% of participants who fall into the depression subscale. The age group between 30 and 39 years is the one that stands out the most with 17.3%, while those over 50 years of age have the lowest values. Additionally, in the anxiety subscale, out of the 40.1% of cases, the 30 to 39 year age group is the most affected (18.8%). This trend continues in the stress subscale where the 30 to 39 year age group represents 18.8% of the total, 10.4% corresponds to the 40 to 49 year age group, and the rest of the cases have lower values according to this data.

The emotional affectation by sex shows that in the depression subscale, 36.6% of the personnel with symptomatology, there is an equal number of men and women, with 18.3%. In the anxiety subscale of 40.1% with symptoms, women represent 21.8% compared to men with 18.3%. In the stress subscale of 38.6% of those affected, men and women present a similar 19.3%.

According to the work area, 36.6% of the sample suffers from depression, with 22.8% corresponding to teachers and to 13.9% administrative personnel. In the anxiety subtest with symptoms, 40.1% and 26.7% correspond to teachers, and 13.4% corresponds to administrative staff. Finally, in the stress subscale of 38.6% with symptoms, 26.7% are teachers and 11.9% are administrative personnel, showing that the group of teachers is the most affected due to the demands of the position.

Furthermore, statistical analysis was conducted using the chi-square test for the case of categorical variables related to depression, anxiety, and stress dimensions. Additionally, we divided our sample into three categories, age range, sex, and work position.

According to age range, we find that this variable is statistically significantly related to depression, anger, and anxiety (Table 3). In the three cases, the level of association measured by Cramer’s V is not high, with a maximum level of 0.17 for the nexus age-anger. On the other hand, in the sex category, men and women, the association was found in stress release (stress) and over-reaction. The Cramer’s V was 0.20 and 0.18, respectively. In the work position category, an association was found for the case of five dimensions: anxiety, depression, over-reaction, difficult to relax, and anxiety (tachycardia). The strength of the association, as seen in the previous categories, is not too strong, but it allows us to validate the hypothesis raised here.

### 3.2. DASS-21 and Productivity Variables

Likewise, according to analysis of the existence of a link between productivity and depression, anxiety, and stress, we carried out an ANOVA test. The results are listed in Table 4.

Three dimensions of anxiety, i.e., dry mouth, skeletal muscle effects, and fear, are statistically significant related to time for easier documentation, one of the four dimensions of productivity. In the case of dry mouth, when analyzing the frequencies, i.e., never, sometimes, frequently, and most of the time, we find that individuals who say they feel anxious sometimes take more time, on average, to perform simple tasks than individuals who say they feel anxiety most of the time. Furthermore, individuals who say they feel anxious frequently take more time, on average, to perform simple tasks than individuals who say they never feel anxiety. Likewise, individuals who say they feel anxious sometimes take more time, on average, to perform simple tasks than individuals who say they never feel anxiety. Finally, individuals who say they feel anxious sometimes take more time, on average, to perform simple tasks than individuals who say they feel anxiety most of the time. In the case of skeletal muscle effects, the results show us that individuals who feel anxiety most of the time take more time, on average, to perform difficult tasks than individuals who say they feel anxiety sometimes. Individuals who feel anxiety most of the time take more time, on average, to perform difficult tasks than individuals who say they never feel anxiety. In addition, individuals who never feel anxiety take more time, on average, to perform difficult tasks than individuals who say they feel anxiety sometimes. In the case of fear, the results show us that individuals who feel anxiety frequently take more time, on average, to perform simple tasks than individuals who say they feel anxiety most of the time. Similarly, individuals who never feel anxiety take more time, on average, to perform simple tasks than individuals who say they feel anxiety sometimes. Moreover, individuals who never feel anxiety take more time, on average, to perform simple tasks than individuals who say they feel anxiety most of the time.

In relation to one dimension of depression, known as dysphoria, individuals who feel depression most of the time are able to generate more documents, on average, than individuals who say they feel depression sometimes. On the other hand, individuals who feel depression most of the time are able to generate more documents on average than individuals who say they never feel depressed. It would be interesting to delve into these results because they are far from what the conventional literature on depression points out, however, that is outside the objective of the present study.

In one of the stress dimensions, specifically in the case of over reactive, the results show us that individuals who overreact sometimes have received more rejections of their documents, on average, than individuals who overreact most of the time. Furthermore, individuals who never overreact have received more rejections of their documents, on average, than individuals who overreact sometimes.

## 4. Discussion

The change in human life caused by the COVID-19 pandemic has been the cause of deaths due to pneumological conditions. However, mental illnesses also appear simultaneously, which could have severe consequences if not identified and diagnosed on time. Despite this, the commercial and economic problems that have been generated have forced workers to return to face-to-face activities. For this reason, the affectation degree presented in teachers and administrative staff was evaluated in this research as a sample taken from the most populated city in Ecuador. The results suggest that the indices of depression, anxiety, and stress symptoms are evident, so it is appropriate to detect whether there are significant changes in the levels of intensity and thus avoid possible disorders that could become chronic in the long term.

The bibliography presented shows that there are already studies where it is proposed to know the levels of anxiety, depression, and stress to which the actors of the educational ecosystem are exposed. Although most studies have focused on the analysis of students, it can be seen that the values obtained are very similar to those of this study. In [7], participants had 33.3% anxiety, 46.92% depression, and 28.5% stress, which coincides with the estimated levels in [9]. Similarly, in another study, 8.25% of the students showed moderate to extreme negative feelings when applying the DASS-21 instrument. Analyzing the mental health status of teachers in the paper by [10], it has been estimated that 32.2% presented symptoms of depression, 49.4% of anxiety, and 50.6% of stress. These results agree with those presented by [11] and with the results of this study.

All of these studies show a trend, and although this instrument has been applied to different samples and geographical sites, the mentioned trend is kept in the research. Therefore, this study evaluates teachers and the administrative staff where the emotional affectation by sex in the three subscales highlights equality in values for both men and women, indicating that this variable does not significantly influence the generation of a mental disorder. In the age analysis, it can be seen that middle-aged people are exposed to high demand due to the constitution of their family responsibilities and their professional training, which is evidenced in the high percentages presented.

The statistical analysis showed a relationship between the socio-demographic variable job position and anxiety-stress variables. When analyzing the activities performed by the administrative personnel, it was found that they required less investment of concentration and time. In contrast, teachers must prepare their classes, grade evaluations, design activities, and invest time in tutoring and support in developing degree projects. This result shows greater demand and pressure in one group than in the other, corroborating the initial hypothesis. When analyzing the depression, anxiety, and stress levels compared to productivity variables, we found that five disorders directly affect productivity variables as stated in the introductory hypothesis.

Authors should discuss the results and how they can be interpreted from the perspective of previous studies and of the working hypotheses. The findings and their implications should be discussed in the broadest context possible. Future research directions may also be highlighted.

## 5. Conclusions

Our study shows that anxiety, stress, and depression have importantly affected both administrative workers and teachers. The impact was differentiated depending on sex, age, and job position. Chi-squared analysis and Cramer’s V effect size showed more notable affectation on sex: sex—stress release and sex—over-reaction and less notably on age. The ANOVA test showed that productivity was affected by anxiety but in a positive way; that is, it would seem that the more anxiety individuals go through, the less time it takes them to perform activities. Likewise, individuals were affected by stress, which increased with the number of rejected documents. As in the case of anxiety, depression apparently allows individuals to generate more documents.

## Figures and Tables

**Table 1 behavsci-12-00420-t001:** Percentage distribution of participants according to sex, age and job position.

Sex	Position	<30 (%)	30 to 39 (%)	40 to 49 (%)	>50 (%)	Total (%)
Men	Administrative	33	10	12	29	18
	Teacher	3	28	42	35	29
Women	Administrative	50	35	19	12	29
	Teacher	13	26	27	24	24

**Table 2 behavsci-12-00420-t002:** Relationship between age, sex, position variables and the presence of disorders.

Socio-Demographic Variables		Depression (n)		Anxiety (n)		Stress (n)	
		Yes	No	Yes	No	Yes	No
Age ranges	≤30	10	14	7	17	8	16
	30–39	35	49	38	46	38	46
	40–49	19	38	25	32	21	36
	≥50	10	27	11	26	11	26
	Total	74	128	81	121	78	124
Sex	Man	37	60	37	60	39	58
	Woman	37	68	44	61	39	66
	Total	74	128	81	121	78	124
Position	Administrative	28	64	27	65	24	68
	Teacher	46	64	54	56	54	56
	Total	74	128	81	121	78	124

**Table 3 behavsci-12-00420-t003:** Significance levels according to the chi-square statistic in the relationship between variables.

Relationship between Variables	Chi-Square Value	Asymptotic Significance Level	Magnitude of Significance (Cramer’s V)
Age—Depression	2.78 **	0.08	0.16
Age—Anger	2.61 *	0.03	0.17
Age—Anxiety (tachycardia)	2.96 **	0.09	0.15
Sex—stress release	4.49 *	0.04	0.20
Sex—over-reaction	1.56 **	0.06	0.18
Position—Anxiety (dry mouth)	5.02 **	0.07	0.19
Position—Depression	10.35 *	0.05	0.20
Position—over-reaction	11.631 *	0.02	0.21
Position—difficult to relax	8.61 **	0.06	0.1
Position—Anxiety (tachycardia)	6.313 **	0.07	0.19

* *p*-value < 0.05; ** *p*-value < 0.1.

**Table 4 behavsci-12-00420-t004:** Significance levels according to the ANOVA statistic in the relationship between variables.

	Productivity Variable				
Statistical Significance	Disorders	Number of Products Made	Number of Products Rejected	Perform Difficult Tasks	Perform Simple Tasks
Kruskal–Wallis	Fear	1.17	0.78	2.96	7.05
*p*-value		0.75	0.85	0.36	0.07 **
Kruskal–Wallis	Anxiety dry mouth	0.83	4.18	10.8	0.8
*p*-value		0.84	0.24	0.01*	0.84
Kruskal–Wallis	Over reactive	5.41	6.05	4.18	1.08
*p*-value		0.15	0.1 **	0.24	0.78
Kruskal–Wallis	Skeletal muscle effects	8.22	0.55	7.83	4.96
*p*-value		0.04 *	0.9	0.05 *	0.174
Kruskal–Wallis	Dysphoria	7.35	0.52	7.37	4.3
*p*-value		0.06 **	0.91	0.06	0.23

* *p*-value < 0.05; ** *p*-value < 0.1.

## Data Availability

Data Availability: http://repositorio.uti.edu.ec//handle/123456789/3709, accessed on 25 January 2022.

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
