# Peer review of "Emotional State of Teachers and University Administrative Staff in the Return to Face-to-Face Mode"

_behavsci, 2022, doi:10.3390/bs12110420_

Round 1
Reviewer 1 Report
Congratulations on the research carried out, as well as the clear presentation of the results. However, the citations in the text are not well indicated, which is why I am attaching the file.
Author Response
Reviewer comments: The citations in the text are not well indicated, which is why I am attaching the file.
All citations have been corrected following the reviewer's recommendations.
Reviewer 2 Report
The contribution is certainly interesting, and the study is well proposed and with adequate numbers, but formally there is an obvious problem with internal citations.
Please note to include square brackets, for example, 2 in [2]. In addition to this, as highlighted in the text it is required to adequately link set and note, for example: According to 5, has no meaning, but instead write According to Arias Gallegos et al., about burnout syndrome [5].
Make the parts marked in yellow in the pdf more narrative and review, minimally, the use of the English language.

Author Response
Reviewer´s comments:
There is an obvious problem with internal citations. Please note to include square brackets, for example, 2 in [2]. In addition to this, as highlighted in the text it is required to adequately link set and note, for example: According to 5, has no meaning, but instead write According to Arias Gallegos et al., about burnout syndrome [5].
All comments have been resolved
Make the parts marked in yellow in the pdf more narrative and review, minimally, the use of the English language.
All comments have been resolved
Reviewer 3 Report
The article is well written, in academic and scientific language, and is an interesting topic for the scientific community. There are no apparent grammatical or spelling errors. There is coherence and quality of the arguments presented by the author(s). The objectives are in clear agreement with the methodology and with the original results and the conclusions provide great knowledge and transference. The title synthesizes the main idea of the paper, is self-explanatory, concise, informative and avoids abbreviations. It also generates reading expectations that are fulfilled. It incorporates the necessary information to guide the reader to identify the basic content of the text quickly and to determine its relevance. It refers, in sufficient quantity and in an adequate manner, to other researches or works carried out in the field of the subject matter addressed. It supports the conceptual reference with sufficient and adequate authority figures. The conclusions are directly derived from the development of the work, and are related to the purpose of the article and the title. Maintains an adequate relationship between the parts: objective (problem, objectives, hypothesis), theoretical framework, methodology, results and conclusions. Citation of mainstream sources (books and journal articles) predominates. References are recent and mostly international.
Author Response
Reviewer's comments:
The reviewer has not proposed any corrections to be made